# Dental findings frequently overlooked in sinus computed tomography reports

**Annina Wuokko-Landén**[1]*, **Hanna Välimaa**[1,2], **Karin Blomgren**[3], **Anni Suomalainen**[4]

1 Department of Oral and Maxillofacial Diseases, Helsinki University Hospital and University of Helsinki, Helsinki, Finland, 2 Meilahti Infectious Diseases and Vaccine Research Center, MeVac, Helsinki University Hospital and University of Helsinki, Helsinki, Finland, 3 Department of Otorhinolaryngology, Helsinki University Hospital and University of Helsinki, Helsinki, Finland, 4 HUS Medical Imaging Center, Department of Radiology, University of Helsinki and Helsinki University Hospital, Helsinki, Finland

* annina.wuokko@fimnet.fi

## Abstract

### Objectives

Computed tomography (CT) and cone beam computed tomography (CBCT) represent the main imaging modalities used in rhinosinusitis patients and are also important in odontogenic sinusitis (OS) diagnostics. Reports, however, often lack information on dentition. Here, we aimed to determine how maxillary dentition is initially interpreted in rhinosinusitis patients' CT/CBCT reports and which dental findings in particular are potentially missed, thus needing more attention.

### Study design

CT/CBCT scans and radiological reports from 300 rhinosinusitis patients were analysed focusing specifically on dental findings. An experienced oral and maxillofacial radiologist re-evaluated the scans and the assessment was compared to the original reports using the McNemar test.

### Results

From the 300 original reports, 233 (77.7%) mentioned the maxillary teeth. The most frequent statement (126/300, 42.0%) was 'no apical periodontitis'. Apical periodontitis and severe alveolar bone loss were significantly overlooked ($p < 0.001$). Amongst the 225 patients for whom the CT/CBCT report initially lacked information on dental pathology, 22 patients were diagnosed with apical periodontitis and 16 with severe alveolar bone loss upon re-evaluation.

### Conclusions

Dental pathology remains underreported in rhinosinusitis patients' CT/CBCT reports. Because these reports affect OS diagnostics, a routine and structured review of the maxillary teeth by a radiologist is necessary. Such examinations should encompass the maxillary teeth.

**Data Availability Statement:** According to Finnish legistation, i.e. The Act on the Secondary Use of Health and Social Data, to protect potentially identifiable data, a separate approval to access data is needed. Data are available from Finnish Social

and Health Data Permit Authority for researchers who meet criteria for access to confidential data. For more information, see www.findata.fi/en/.

**Funding:** Statistical analysis of this study was financially supported by the Helsinki University Hospital Research Fund (grant no. Y1014SU011/1149010/30101). The funders had no role in study design, data collection and analysis, decision to publish, or preparation of the manuscript.

**Competing interests:** The authors have declared that no competing interests exist.

## Introduction

Odontogenic sinusitis (OS) is a common, often overlooked disease [1]. Computed tomography (CT) plays a key role in revealing OS [2] and remains the method of choice as an imaging modality in chronic rhinosinusitis patients [3]. Cone beam computed tomography (CBCT), featuring a lower radiation dose, but a poorer view of soft tissues, can serve as an alternative [4]. In addition, CT and CBCT provide valuable data on the adjacent dentition and maxillary bone.

Radiological reports from rhinosinusitis patients frequently exclude dental pathology [5], and otolaryngologists and rhinologists have stated that radiologists rarely comment on dental pathology in CT scans [6]. Insufficient radiological reports may mislead the diagnosis and delay appropriate treatment and recovery. Untreated OS causes burdens to healthcare systems and can lead to persistent symptoms or even life-threatening intra-cranial complications.

Various dental conditions and procedures can lead to OS [7]. Apical periodontitis [8] and alveolar maxillary bone loss [9] associate with sinus mucosal thickening. Moreover, sinonasal complications often result from dental treatment, such as tooth extractions, endodontics, and implant surgery including alveolar ridge augmentation procedures [10]. Occasionally, maxillary sinus disorders are linked to peri-implantitis, as well as ectopic teeth and related dentigerous cysts [11]. In the case of a rhinosinusitis patient, most dental pathologies must be considered as possible causative agents.

Effective treatment protocols and several successful case series on managing OS patients have been published [10, 12]. Utilising these, however, calls for determining the correct diagnosis and the specific dental cause of disease.

Here, we aimed to investigate (i) how often and (ii) with which precision dental pathology is evaluated in the original CT/CBCT reports of rhinosinusitis patients and (iii) whether some findings are overlooked. Additionally, we studied how often an insufficient area of interest (AOI) or artefacts hinder the accurate interpretation of radiological findings concerning teeth. Our primary aim was to improve OS radiological diagnostics and identify areas needing more attention.

## Materials and methods

Data were retrospectively obtained for patients who visited a tertiary hospital, namely, Helsinki University Hospital's (HUH) Department of Otorhinolaryngology, in 2013 because of acute or chronic rhinosinusitis.

The records of all 2366 patients with International Statistical Classification of Diseases and Related Health Problems tenth revision (ICD-10) code J32 (chronic sinusitis) or J01 (acute sinusitis) including subcodes for maxillary, frontal, sphenoidal, ethmoidal, pansinusitis, other, and unspecified sinusitis were first scrutinised. Patients with clearly isolated frontal sinusitis and sphenoidal sinusitis were excluded because of their unlikely association with odontogenic factors. In addition, patients with previous visits to HUH before 2013 because of rhinosinusitis or with a known history or clear radiological proof of sinus surgery were excluded. From this patient cohort we included patients aged >18 years with CT/CBCT examination and its original report electronically available. Fig 1 illustrates the patient selection. CT/CBCT images and their original reports were used as the study subjects.

Radiological examination was performed either at the referring clinic or in our tertiary healthcare setting. Examinations were originally interpreted by a specialist in radiology or a resident. Following interpretation by a resident, a specialist had confirmed the report.

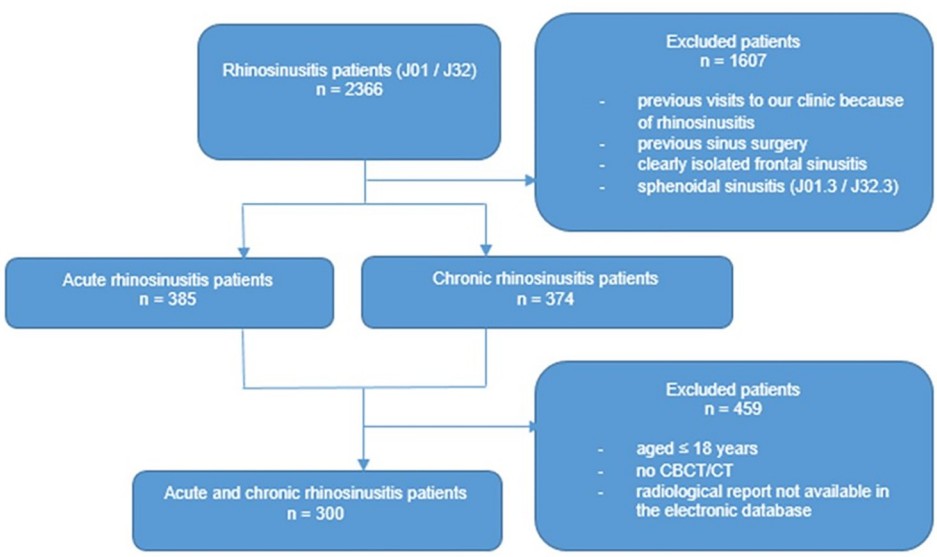

**Fig 1. Flowchart of patient selection.**

## Data collection

All comments on the maxillary teeth and dental pathology were obtained from the original reports. Findings were categorised per tooth as precisely as possible. We counted reports in which OS was suspected or excluded.

An oral and maxillofacial (OMF) radiologist with more than 20 years of clinical experience and blinded to the original reports re-evaluated all CT/CBCT scans. The images were analysed on a Barco E-3620 3MP Medical Flat Grayscale Display (Barco, Kortrijk, Belgium) with a diagonal dimension of 20.8 inches and a resolution of 2.048 × 1.536 pixels with a pixel size of 0.207 mm.

Table 1 lists the dental findings the OMF radiologist sought.

We compared the most typical OS dental findings from the original CT/CBCT reports and the OMF radiologist's re-evaluation and determined the level of agreement between these findings (Table 2).

## Statistical analyses

Statistical analyses were performed using SAS System for Windows, version 9.4 (SAS Institute Inc., Cary, NC, USA). Data are described using the mean ± standard deviation (SD) for continuous variables and using frequencies and percentages for categorical variables. The McNemar test was used to assess the differences in the number of dental findings between the original reports and the OMF radiologist's re-evaluation. Agreement between the original reports and the OMF radiologist's re-evaluation was calculated using the kappa coefficients. Kappa values were classified as <0 indicating no agreement, 0–0.20 as slight agreement, 0.21–0.40 as fair agreement, 0.41–0.60 as moderate agreement, 0.61–0.80 as good agreement, and 0.81–1 as very good agreement. We considered $p < 0.05$ as statistically significant.

## Ethical approval

The study protocol was approved by HUH. No Research Ethics Board review was required. All components of this study complied with Finnish legislation and follow the Helsinki

**Table 1. Dental findings assessed by an oral and maxillofacial radiologist.**

| |
|---|
| 1. Apical periodontitis (either perforating or not perforating the sinus floor) |
| 2. Marginal alveolar bone loss (extending at least to the middle third of the root or a furcation lesion in a multirooted tooth to the same level) |
| 3. Root remnant (inside bone or perforating the sinus or oral cavity, with or without signs of infection) |
| 4. Pulp-perforating caries |
| 5. Root fracture |
| 6. Suspected radicular cyst (periapical lesion at least 10 mm in size) |
| 7. Supernumerary teeth (retained or erupted) |
| 8. Lateral (canal) infection |
| 9. Root canal treated tooth |
| 10. Inadequate root canal treatment (underfilled, untreated, or overfilled root canals) |
| 11. Oro-antral communication |
| 12. Dental implant |
| 13. Peri-implantitis |
| 14. Amputated pulp |
| 15. Foreign body in the maxillary sinus |
| 16. Edentulous maxilla |
| 17. Extracted teeth |
| 18. Wisdom tooth (extracted, erupted, or partially erupted) |

Declaration. Due to the retrospective nature of the study, no informed consents were needed. The data were first accessed 02-12-13 and later during the years 2014–2019, with patient confidentiality ascertained in each step of the data collecting, maintaining, and processing.

## Results

In total, 296 CT and 4 CBCT images with the original interpretations were available for re-evaluation. The rhinosinusitis patient cohort consisted of 196 female (65.3%) and 104 male (34.7%) patients with a mean age of 45.2 ± 16.7 years.

### Original radiological reports and their dental findings

Teeth were mentioned in 233 reports (77.7%), and in 126 cases (42.0%) the phrase related to dentition was 'no apical periodontitis'. Altogether, 225 reports (75.0%) mentioned neither maxillary teeth nor dental pathology. In addition, 8 of 300 cases (2.7%) were apparent OS and the possibility of OS was excluded in 3 reports. In 1 report, consulting a dentist was recommended. Furthermore, 14 reports mentioned 'no deep infectious lesions in the maxillary teeth' and 3 reports included the statements 'nothing special in the maxillary teeth', 'nothing acute in the maxillary teeth', and -nothing to mention about the maxillary teeth'. We classified these reports as having no apical periodontitis.

Other dental findings reported consisted of an amputated right maxillary molar (1 patient), incisor pathology such as apical periodontitis (9 patients), nonpathological findings such as persisting milk tooth canine and edentulous maxillary jaw (13 patients), and dental implants (4 patients).

In 36 cases an OMF radiologist was consulted or had originally interpreted the examination. A total of 30 radiologists or residents participated in the initial CT/CBCT interpretations. Radiologists' experiences varied from being a resident to being a specialist in radiology with over 30 years of clinical experience.

**Table 2. Dental findings from computed and cone beam computed tomographies of rhinosinusitis patients (n = 300).** Comparison of original interpretations and oral and maxillofacial (OMF) radiologist's re-evaluation.

| Dental finding in canine–molar region | Original report | | Re-evaluation by OMF radiologist | | p value | κ coefficient |
|---|---|---|---|---|---|---|
| | n | % | n | % | | |
| Apical periodontitis[a] | 46 | 15.3 | 70 | 23.3 | <0.001 | 0.661 |
| Perforating the sinus floor | 9 | 3.0 | 19 | 6.3 | 0.008 | 0.479 |
| Not perforating the sinus floor | 38 | 12.7 | 56 | 18.7 | 0.002 | 0.574 |
| Marginal alveolar bone loss | 8[b] | 2.7 | NA | NA | <0.001 | 0.103 |
| Severe[c] | 3 | 1.0 | 33 | 11.0 | | |
| Undefined extent or location | 5 | 1.7 | NA | NA | | |
| Caries | 6[b] | 2.0 | NA | NA | 0.132 | 0.336 |
| Pulp perforating | 2 | 0.7 | 11 | 2.7 | | |
| Undefined depth | 4 | 1.3 | NA | NA | | |
| Root canal overfilling | 3 | 1.0 | 11 | 3.7 | 0.005 | 0.419 |
| Root canal underfilling[d] | 2 | 0.7 | 31 | 10.3 | <0.001 | 0.049 |
| Untreated root canal | 1 | 0.3 | 27 | 9.0 | <0.001 | 0.065 |
| Oro-antral communication | 3 | 1.0 | 6 | 2.0 | 0.180 | 0.437 |
| Suspected radicular cyst | 3 | 1.0 | 4 | 1.3 | 0.333 | 0.566 |
| Foreign body in the maxillary sinus | 2 | 0.7 | 1 | 0.3 | 0.564 | -0.0045 |
| Root fracture | 2 | 0.7 | 5 | 1.7 | 0.083 | 0.567 |

n = number of patients.

[a] Periodontal space at least twice the normal width and seen in at least two projections (only in the re-evaluation).

[b] Value used in the statistical analysis.

[c] Alveolar bone loss extending at least to the middle third of the root or furcation lesion in a multirooted tooth to the same level.

[d] Distance from the root canal filling ≥2 mm to the apical foramen.

## Dental findings in the OMF radiologist's re-evaluation

According to the OMF radiologist's re-evaluation, among 225 examinations originally lacking any reported dental pathology, 22 had apical periodontitis, 16 had severe alveolar bone loss, 4 had pulp perforating caries, 16 had underfilled root canals, 11 had unfilled root canals, 2 had overfilled root canals, and 1 had a fractured root.

Table 2 summarises the dental findings commonly associated with OS and compares these with the OMF radiologist's re-evaluation. Figs 2 and 3 provide examples of dental findings not mentioned in the original reports.

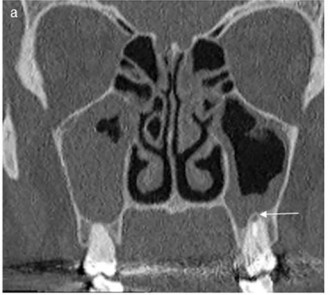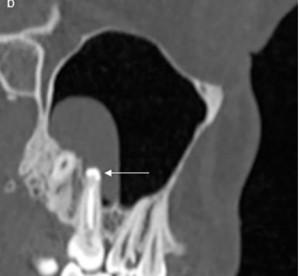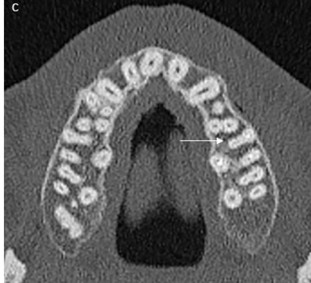

**Fig 2.** Overlooked dental findings in maxillary molars (marked with arrows) from computed tomography scans: (a) coronal view of an apical periodontitis; (b) sagittal view of a root canal overfilling in a palatal root generally regarded as harmless, mesiocentral root canal with a suboptimal root canal filling with apical periodontitis (not shown) and a suspected reactive retention cyst; and (c) axial view of an untreated mesiocentral root canal in a mesiobuccal root.

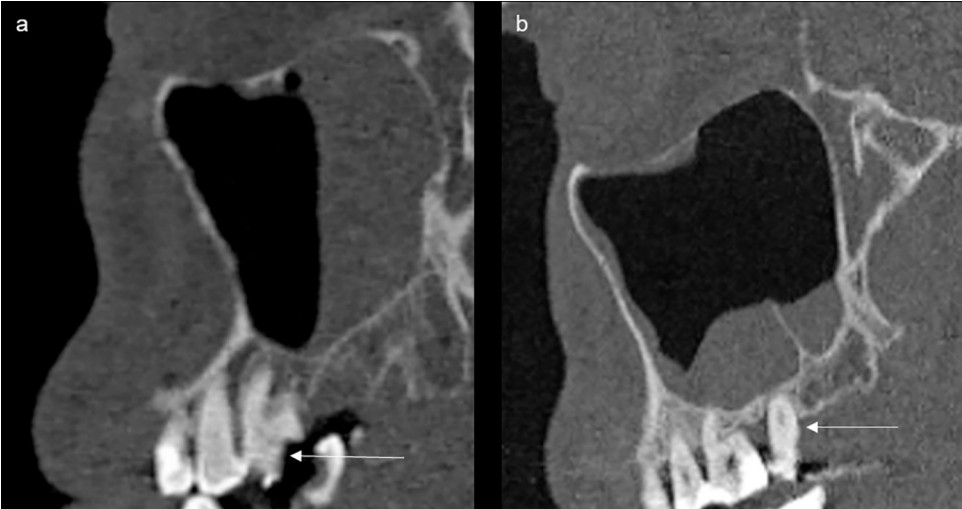

**Fig 3.** Overlooked dental findings in maxillary molars (marked with arrows) from computed tomography scans: (a) sagittal view of a pulp perforating caries and (b) sagittal view of severe marginal alveolar bone loss.

In the canine–molar region, dental pathology was identified in 102 patients (34.0%) with apical periodontitis emerging as the most common dental pathology. At least 1 root canal treated maxillary tooth was found in 88 patients (29.3%), among whom 27 (9.0%) had at least 1 untreated root canal. Apical periodontitis, marginal alveolar bone loss, and root canal treatment deficiencies typically related to the molars.

Because of the technical limitations of the CT/CBCT scans, diagnosing caries was not feasible on both sides for 219 patients. Overall, 18% of the examinations covered maxillary tooth crowns and marginal alveolar bone. These structures were either partly or not at all visible in the remainder of the examinations. Almost one-third (31.0%) had an artefact that hindered evaluation of the dental pathology. Moreover, 145 examinations (48.3%) had no artefact, but their AOI was inadequate to diagnose the crown and marginal alveolar bone.

Finally, 2 patients had a root remnant perforating the oral cavity and 1 patient had an infected root remnant inside the maxillary bone.

## Discussion

In our cohort of rhinosinusitis patients' CT/CBCT images and their original radiological reports, information about dentition was infrequently and briefly reported, largely restricted to apical periodontitis. Furthermore, among 225 patients whose original radiological reports did not mention dental pathology, 22 patients had apical periodontitis, 16 had severe alveolar bone loss, and 4 had pulp perforating caries according to the OMF radiologist. Agreement on sinus perforating apical periodontitis and non-perforating apical periodontitis were only moderate. These findings show that important radiological dental findings possibly causing sinonasal symptoms can be easily overlooked.

Apical periodontitis was the most frequent dental pathology, present in approximately one-fifth of the patients. The majority of the lesions did not perforate the sinus floor radiologically, rendering assessment of their possible role in sinus symptoms more challenging. Infection originating from apical periodontitis can actually advance into the sinus through the maxillary bone via the lymph and blood vessels without any bone destruction occurring [13]. Therefore, a radiologically intact cortex can be found between the apical lesion and maxillary sinus in

cases of OS, and the affected tooth should be evaluated clinically by a dentist. CT provides valuable information also for the dental procedure.

In almost half (48.3%) of the images, AOI did not cover the tooth crown and marginal alveolar bone for diagnostics. According to the OMF radiologist, severe alveolar bone loss was, nevertheless, visible in 11.0% of the images, and thus originally significantly underreported. Maxillary sinus mucosal thickening associates with moderate and severe alveolar bone loss [9], and periodontitis associates with chronic rhinosinusitis [14]. Therefore, severe alveolar bone loss should also be seriously considered and treated as a possible condition underlying a patient's sinus symptoms.

Artefacts hindered the interpretation of dental findings in 31.0% of cases. One artefact in CT/CBCT is a distortion or error in an image unrelated to the studied subject, which significantly affects the evaluation of pathologies such as caries [15]. Notably, pulp-extending caries can cause sinus mucosal changes before radiologically visible apical periodontitis is observed. Hence, as artefact may limit the utility of CT/CBCT for dental pathology, also 2-dimensional images have a role for OS diagnostics when pulpal necrosis is suspected.

As expected, the pathological dental findings in this study most often related to the maxillary molars. Compared to the premolars, the maxillary molars often result in root canal treatment failures [16]. Additionally, the molar roots accompany the shortest distance to the maxillary sinus floor. The molars are also most likely to have apical bone defects [17] and develop periodontitis. Teeth with an inadequate root canal filling are more likely to present with apical periodontitis [18]. In our study, at least one root canal treated maxillary tooth was found in 29.3% of patients, among whom almost one-third had at least 1 untreated root canal. Although root canal overfilling does not directly cause rhinosinusitis, resistance to infections may be weakened because of changes in the sinus mucosa.

The main characteristic of the reports studied was the sparse comment on the maxillary teeth. According to the OMF radiologist, dental pathology was observed in over one-third of the patients in the maxillary canine–molar region. In original reports, OS was apparent according to a radiologist in 8 of 300 (2.7%), excluded in 3 of 300 cases, and only 1 report from 300 recommended consulting a dentist. Considering the relevance of the possible dental finding was thus primarily left to the ear, nose, and throat specialist in question.

A standardised and structured format improves the quality of radiological reporting [19]. Furthermore, attention should be paid to using accurate terminology. A more thorough report of dental findings could simplify the process of determining the treatment path. We have previously suggested that dental findings could be primarily categorised as follows: (1) radiological findings referring to OS, (2) potential radiological findings referring to OS, and (3) no radiological findings referring to OS. Consulting a dentist should always be included in the first two categories [20]. In addition, the consulting dentist should have access to all relevant radiological examinations.

The strength of this study lies in the usability of these results to improve OS diagnostics and clinical practice guidelines [21]. Some limitations to this study should also be acknowledged. First, re-evaluation was performed by a single OMF radiologist in a single centre. Second, because of the retrospective nature of our study and our inability to verify the diagnosis, we could not categorise pathological dental findings as an obvious or potential cause of OS nor as directly causing sinonasal symptoms.

## Conclusions

In relation to dental findings, computed tomography and cone beam computed tomography reports for rhinosinusitis patients remained limited and cursory, whereby alveolar bone loss

was particularly overlooked. In our study, both apical periodontitis and severe alveolar bone loss were originally significantly underdiagnosed. The radiologist's contribution to odontogenic sinusitis diagnostics or reporting alleged pathological dental findings as a cause of sinonasal symptoms–or alternatively as only co-existing findings—is essential. Therefore, comments on dentition visible in the optimised area of interest should always be included in a sinus computed tomography report.

## Acknowledgments

Statistical analysis was performed by statistician Tero Vahlberg.

## Author Contributions

**Conceptualization:** Annina Wuokko-Landén, Hanna Välimaa, Karin Blomgren, Anni Suomalainen.

**Data curation:** Annina Wuokko-Landén, Anni Suomalainen.

**Formal analysis:** Annina Wuokko-Landén.

**Funding acquisition:** Karin Blomgren.

**Investigation:** Annina Wuokko-Landén, Karin Blomgren, Anni Suomalainen.

**Methodology:** Annina Wuokko-Landén, Hanna Välimaa, Karin Blomgren, Anni Suomalainen.

**Project administration:** Hanna Välimaa, Karin Blomgren.

**Resources:** Hanna Välimaa, Karin Blomgren.

**Supervision:** Hanna Välimaa, Karin Blomgren.

**Validation:** Annina Wuokko-Landén, Anni Suomalainen.

**Visualization:** Annina Wuokko-Landén, Anni Suomalainen.

**Writing – original draft:** Annina Wuokko-Landén.

**Writing – review & editing:** Annina Wuokko-Landén, Hanna Välimaa, Karin Blomgren, Anni Suomalainen.

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
