## [Decision Letter · Decision Letter 0]

11 Jan 2024

PONE-D-23-33980Dental findings frequently overlooked in sinus computed tomography reportsPLOS ONE

Dear Dr. Wuokko-Landén,

Thank you for submitting your manuscript to PLOS ONE. After careful consideration, we feel that it has merit but does not fully meet PLOS ONE’s publication criteria as it currently stands. Therefore, we invite you to submit a revised version of the manuscript that addresses the points raised during the review process.

Dear Authors,

The authors are advised to read the comments carefully and carry out the changes accordingly.

We look forward to receiving your revised manuscript.

Kind regards,

Mohmed Isaqali Karobari, BDS, MScD.Endo, MFDS.RCPS Glasg, Ph.D. scholar

Academic Editor

PLOS ONE

Journal Requirements:

"This study was supported by the Helsinki University Hospital Research Fund (statistical analysis). "

5. We note that Figures 2 and 3 in your submission contain copyrighted images. All PLOS content is published under the Creative Commons Attribution License (CC BY 4.0), which means that the manuscript, images, and Supporting Information files will be freely available online, and any third party is permitted to access, download, copy, distribute, and use these materials in any way, even commercially, with proper attribution. For more information, see our copyright guidelines: http://journals.plos.org/plosone/s/licenses-and-copyright.

a. You may seek permission from the original copyright holder of Figures 2 and 3 to publish the content specifically under the CC BY 4.0 license. 

Reviewers' comments:

Reviewer's Responses to Questions

**Comments to the Author**

1. Is the manuscript technically sound, and do the data support the conclusions?

Reviewer #1: No

Reviewer #2: Yes

Reviewer #3: Yes

2. Has the statistical analysis been performed appropriately and rigorously? 

Reviewer #1: No

Reviewer #2: Yes

Reviewer #3: Yes

3. Have the authors made all data underlying the findings in their manuscript fully available?

Reviewer #1: Yes

Reviewer #2: Yes

Reviewer #3: Yes

4. Is the manuscript presented in an intelligible fashion and written in standard English?

Reviewer #1: Yes

Reviewer #2: Yes

Reviewer #3: Yes

5. Review Comments to the Author

Reviewer #1: Even though the study examines how maxillary dentition is interpreted in CT/CBCT reports of rhinosinusitis patients, it does not meet rigorous academic standards as it solely relies on observational research and does not establish any statistical correlations. The absence of statistical analyses reduces the study's strength and makes it vulnerable to subjective interpretations. Although the study addresses the significance of dentition information in rhinosinusitis imaging, it lacks a quantitative approach, thereby limiting its ability to contribute to evidence-based knowledge. By incorporating statistical correlations into a more comprehensive methodology, the academic credibility and reliability of the findings would be enhanced.

Reviewer #2: Dear Authors,

In general, the idea of this study, regarding the investigation of how often and with which precision dental pathology is evaluated in the original CT/CBCT reports of rhinosinusitis patients, whether some findings are overlooked, and how often an insufficient area of interest or artefacts hindered the accurate interpretation of radiological findings concerning teeth is interesting.

The role of these aspects in medicine needs further studies that could open a creative matter of debate in literature by adding new information.

The study was well conducted by the authors; however, there are some concerns to revise that are described below:

- Better formulate the abstract section describing the aim of the study.

- The introduction section resumes the existing knowledge regarding this topic but at the end of this section, Authors should underline the rationale of the study.

- In the central section, Authors should better clarify the inclusion and exclusion criteria.

- Better describe how the sample size of the study was calculated.

- The discussion section appears well organized. Please add a specific sentence that clarifies the results obtained in the first part of the discussion.

- The conclusion should reinforce in light of the discussions.

- Add the strenght of the study.

Best regards.

Reviewer #3: Manuscript reviewed: Dental findings frequently overlooked in sinus computed tomography reports

Type: Research Article

Ref: Submission: PONE-D-23-33980

Reviewer name: Dr Sushma Bommanavar

Brief Summary of the Manuscript reviewed: the original study is an exploratory study aimed to determine how maxillary dentition is initially interpreted in rhinosinusitis patients’ CT/CBCT reports and which findings in particular are potentially missed.

Overall impression of the work, noting both strengths and weaknesses: The strength lies in exploratory hypothesis of finding out the potentially missed area in interpretation of CT/CBCT reports among rhinosinusitis patients. The weakness lies in consideration of only one pathology for interpretation. However, this can be quiet minimal weakness as the study highlights the strength part more as diagnostic accuracy studies that can help in future clinical practices.

A good study design & well written but needs few modifications to be made.

Introduction: Well Written

Methodology: Well Written

Result: Well Written

Discussion: Well Written

The original article can be considered if the following revisions are done:

MAJOR ISSUES:

1. Kindly follow the AGREE checklist (BMJ 2016;352:i1152 ) & STRAD checklist, if applicable

MINOR ISSUES:

1. In figure legend description for figure no 2 & 3, kindly reduce the total word count.

6. PLOS authors have the option to publish the peer review history of their article (what does this mean?). If published, this will include your full peer review and any attached files.

Reviewer #1: No

Reviewer #2: **Yes: **Cinzia Maspero

Reviewer #3: No

---

## [Author Response · Author response to Decision Letter 0]

30 Jan 2024

Dear Dr Mohmed Isaqali Karobari,

Thank you for your letter and for the reviewers’ encouraging and valuable comments on our manuscript entitled “Dental findings frequently overlooked in sinus computed tomography reports”. All of these comments were very helpful for revising and improving our paper. We have studied these comments carefully and have made corresponding corrections that we hope will meet with your approval. The changes in the revised manuscript are marked by Track changes. In addition, we provide a point-by-point response to the reviewers’ comments below. We hope that the manuscript is now acceptable for publication in its present form. If you have any further queries, please do not hesitate to contact us.

Kind regards,

Annina Wuokko-Landén DDS

annina.wuokko@fimnet.fi

Journal Requirements:

1. Response: 

We thank you for providing the additional information. We have checked the manuscript to ensure that it adheres to the style requirements, and have made some adjustments to the text headings (bolding and italics) as necessary. 

2. Response:

We have now included the grant number (grant number is Y1014SU011/1149010/30101) to the Disclosure statement as requested (see pg 1). 

"This study was supported by the Helsinki University Hospital Research Fund (statistical analysis)."

3. Response:

The cover letter and title page have been revised to include the following statement (see pg 1): 

‘The funders had no role in the study design, data collection and analysis, decision to publish, or the preparation of the manuscript.’ The funding disclosure statement was changed to ‘Statistical analysis of this study was financially supported by the Helsinki University Hospital Research Fund (grant no. Y1014SU011/1149010/30101).’

4. Response:

All relevant data are provided within the manuscript. Due to the sensitive patient information included in our dataset, the raw data cannot be shared via public repositories. 

5. We note that Figures 2 and 3 in your submission contain copyrighted images. All PLOS content is published under the Creative Commons Attribution License (CC BY 4.0), which means that the manuscript, images, and Supporting Information files will be freely available online, and any third party is permitted to access, download, copy, distribute, and use these materials in any way, even commercially, with proper attribution. For more information, see our copyright guidelines: http://journals.plos.org/plosone/s/licenses-and-copyright.

a. You may seek permission from the original copyright holder of Figures 2 and 3 to publish the content specifically under the CC BY 4.0 license. 

Response:

All three of the submitted figures were created by our group from our own dataset. 

Response to the Reviewers:

Reviewer #1: 

Even though the study examines how maxillary dentition is interpreted in CT/CBCT reports of rhinosinusitis patients, it does not meet rigorous academic standards as it solely relies on observational research and does not establish any statistical correlations. The absence of statistical analyses reduces the study's strength and makes it vulnerable to subjective interpretations. Although the study addresses the significance of dentition information in rhinosinusitis imaging, it lacks a quantitative approach, thereby limiting its ability to contribute to evidence-based knowledge. By incorporating statistical correlations into a more comprehensive methodology, the academic credibility and reliability of the findings would be enhanced.

Response:

We thank the reviewer for their careful reading of our manuscript and for their useful and helpful comments on improving our manuscript.

In the Materials and methods section, we have described the statistical methods we employed. The McNemar test was used to compare the originally reported CT/CBCT radiological dental findings with a comprehensive re-evaluation of the images by an oral and maxillofacial radiologist. Kappa coefficients were also calculated to determine the level of agreement between the original reports and the re-evaluation of the scans. We have revised the abstract based on this comment (see pg 2). 

Reviewer #2: 

Dear Authors,

In general, the idea of this study, regarding the investigation of how often and with which precision dental pathology is evaluated in the original CT/CBCT reports of rhinosinusitis patients, whether some findings are overlooked, and how often an insufficient area of interest or artefacts hindered the accurate interpretation of radiological findings concerning teeth is interesting.

The role of these aspects in medicine needs further studies that could open a creative matter of debate in literature by adding new information.

The study was well conducted by the authors; however, there are some concerns to revise that are described below:

- Better formulate the abstract section describing the aim of the study.

- The introduction section resumes the existing knowledge regarding this topic but at the end of this section, Authors should underline the rationale of the study.

- In the central section, Authors should better clarify the inclusion and exclusion criteria.

- Better describe how the sample size of the study was calculated.

- The discussion section appears well organized. Please add a specific sentence that clarifies the results obtained in the first part of the discussion.

- The conclusion should reinforce in light of the discussions.

- Add the strenght of the study.

Best regards.

Response:

We thank the reviewer for their careful reading of our manuscript and for their useful and helpful comments on improving our manuscript. We have adjusted the text as follows:

- The Abstract now defines the aim of the study (see pg 2).

- The rationale for this study is now underlined in the Introduction (see pg 3).

- To clarify the inclusion and exclusion criteria, the Materials and methods section has been revised (see pg 4).

- The sample size (n = 300) represents the total number from 759 adult acute/chronic rhinosinusitis patients with a CT/CBCT image and for whom the original report was available. All patients visited our hospital during the same year (see pg 4). 

- One sentence is added to the first section of the Discussion section (see pg 10). 

- We have slightly modified the Conclusions section (see pp 12–13) and added the strengths of our study to the Discussion section (see pg 12). 

Reviewer #3: 

Manuscript reviewed: Dental findings frequently overlooked in sinus computed tomography reports

Type: Research Article

Ref: Submission: PONE-D-23-33980

Reviewer name: Dr Sushma Bommanavar

Brief Summary of the Manuscript reviewed: the original study is an exploratory study aimed to determine how maxillary dentition is initially interpreted in rhinosinusitis patients’ CT/CBCT reports and which findings in particular are potentially missed.

Overall impression of the work, noting both strengths and weaknesses: The strength lies in exploratory hypothesis of finding out the potentially missed area in interpretation of CT/CBCT reports among rhinosinusitis patients. The weakness lies in consideration of only one pathology for interpretation. However, this can be quiet minimal weakness as the study highlights the strength part more as diagnostic accuracy studies that can help in future clinical practices.

A good study design & well written but needs few modifications to be made.

Introduction: Well Written

Methodology: Well Written

Result: Well Written

Discussion: Well Written

The original article can be considered if the following revisions are done:

MAJOR ISSUES:

1. Kindly follow the AGREE checklist (BMJ 2016;352:i1152 ) & STRAD checklist, if applicable

MINOR ISSUES:

1. In figure legend description for figure no 2 & 3, kindly reduce the total word count.

Response:

We thank you for your careful reading of our manuscript and the helpful suggestions for improving it. We have looked into the AGREE checklist and consider it a good tool for future clinical guidelines of odontogenic sinusitis diagnostics and further research. As such, we have added it to the references (see pg 12). As we mentioned in the Discussion section, one limitation of our retrospective study is the inability to verify the possible diagnosis and our inability to categorise pathological dental findings as an obvious potential cause of OS nor as directly causing sinonasal symptoms. Therefore, our study is not precisely a diagnostic accuracy study and, thus, not all STARD checklist points can be fulfilled. 

We have condensed the legends for Figures 2 and 3 and reduced the overall word count.

---

## [Decision Letter · Decision Letter 1]

12 Feb 2024

Dental findings frequently overlooked in sinus computed tomography reports

PONE-D-23-33980R1

Dear Dr. Wuokko-Landén,

We’re pleased to inform you that your manuscript has been judged scientifically suitable for publication and will be formally accepted for publication once it meets all outstanding technical requirements.

Kind regards,

Mohmed Isaqali Karobari, BDS, MScD.Endo, MFDS.RCPS Glasg, Ph.D. scholar

Academic Editor

PLOS ONE

Additional Editor Comments (optional):

Dear Authors,

The authors have addressed all the comments and revised the manuscript accordingly. The manuscript has greatly improved and can be accepted for publication. I would like to congratulate the authors and wish them all the very best for their future endeavors.

Best regards and keep well

Reviewers' comments:

Reviewer's Responses to Questions

**Comments to the Author**

1. If the authors have adequately addressed your comments raised in a previous round of review and you feel that this manuscript is now acceptable for publication, you may indicate that here to bypass the “Comments to the Author” section, enter your conflict of interest statement in the “Confidential to Editor” section, and submit your "Accept" recommendation.

Reviewer #2: All comments have been addressed

Reviewer #3: All comments have been addressed

2. Is the manuscript technically sound, and do the data support the conclusions?

Reviewer #2: Yes

Reviewer #3: Yes

3. Has the statistical analysis been performed appropriately and rigorously? 

Reviewer #2: Yes

Reviewer #3: Yes

4. Have the authors made all data underlying the findings in their manuscript fully available?

Reviewer #2: Yes

Reviewer #3: Yes

5. Is the manuscript presented in an intelligible fashion and written in standard English?

Reviewer #2: Yes

Reviewer #3: Yes

6. Review Comments to the Author

Reviewer #2: Dear Authors

all my requirements have been addressed. Now the manuscript has been improved.

I suggest to accept it.

Best regards

Reviewer #3: All comments have been well addressed and the manuscript can be accepted for future proceedings.

All comments have been well addressed and the manuscript can be accepted for future proceedings.

7. PLOS authors have the option to publish the peer review history of their article (what does this mean?). If published, this will include your full peer review and any attached files.

Reviewer #2: **Yes: **Cinzia Maspero

Reviewer #3: No
